# Understanding Pharmacy Students’ Preparedness towards Counseling over Cannabis Use Disorder

**DOI:** 10.3390/pharmacy12030077

**Published:** 2024-05-13

**Authors:** Sourab Ganna, Jerusha Daggolu, Sujit S. Sansgiry

**Affiliations:** Department of Pharmaceutical Health Outcomes and Policy, College of Pharmacy, University of Houston, Houston, TX 77204, USA; sganna@cougarnet.uh.edu (S.G.); jdaggolu@cougarnet.uh.edu (J.D.)

**Keywords:** pharmacy education, counseling, curriculum development, student development, cannabis education

## Abstract

The rise in cannabis use prompts significant concerns regarding pharmacy students’ abilities to counsel patients over cannabis use disorder. This study aims to understand pharmacy students’ preparedness to counsel patients with cannabis use disorder (CUD) and evaluate the relationship between knowledge, attitudes towards medical cannabis (MC) and recreational cannabis (RC), and behavior intention (BI) to counsel over CUD. A cross-sectional survey was administered to pharmacy students. Descriptive analyses of sample characteristics were assessed with the *t*-test and one-way ANOVA test. Pearson correlation and linear regression were conducted, measuring the strength and direction of relationships. The average scores for knowledge, attitudes towards MC use and RC, and behavioral intention were 81% (SD 16%), 4.13 (SD 0.75), 3.28 (0.80), and 2.74 (1.00). Significant correlations were observed between knowledge–attitudes toward MC, knowledge–attitudes towards RC, and attitudes towards RC–behavioral intentions. Linear regression indicated attitudes towards MC use and RC, academic year, awareness of MC use legality, obtained knowledge, and past patient interaction were significantly associated with behavioral intention on confidence in counseling over CUD. There is a gap in students’ behavioral intention to counsel. These findings emphasize the importance of ample preparation that enables student pharmacists to address patient needs related to cannabis use confidently.

## 1. Introduction

Cannabis, commonly known as marijuana, is a plant used for medicinal, recreational, and industrial purposes. Containing numerous active compounds, the most well-known being delta-9-tetrahydrocannabinol (THC), along with cannabidiol (CBD). Cannabis has gained significant attention recently due to its well-documented therapeutic benefits in managing various medical conditions, such as chronic pain, epilepsy, multiple sclerosis, and chemotherapy-related nausea [1,2,3]. However, the perception of cannabis within the medical community has been characterized by a complex mix of uncertainty, controversy, and optimism [4,5]. While many advocate for its use medically as a promising treatment option, others are more skeptical and emphasize the need for rigorous scientific evidence to support its effectiveness and safety [6]. As the landscape of cannabis-based medicine continues to evolve, it becomes more essential for healthcare providers, especially future pharmacists, to stay informed about the latest research and evidence-based guidelines to make informed patient decisions. 

As much as 147 million, or 2.5%, of the world’s population have used RC annually at least once, compared to only 0.2% consuming cocaine and 0.2% consuming opiates [7]. In the United States, RC is the most used federally illegal drug, with approximately 48.2 million people, or about 18%, users in 2019 [8,9]. About 30% of those using cannabis have cannabis use disorder [8]. For people who begin using cannabis before age 18, the risk of developing cannabis use disorder is even more significant [10]. As per the *Diagnostic and Statistical Manual of Mental Disorders* (*DSM–5*), CUD is defined as a problematic pattern of cannabis use leading to clinically significant impairment or distress followed by at least two other symptoms within 12 months [11]. Long-term or frequent cannabis use has been linked to an increased risk of psychosis or schizophrenia in some users [12,13].

Changes in state and federal legislation, practice guidelines, and facility policies may pose new challenges for pharmacists in addressing issues associated with cannabis use. Current and future pharmacists are critical in ensuring safe and effective medication management, especially with increased cannabis use. With MC gaining prominence, pharmacy curriculums nationwide may need to prepare students with the necessary knowledge and skills to handle CUD interventions. This includes understanding the pharmacology of cannabis and its cannabinoids, potential drug interactions, dosing guidelines, and counseling patients on its proper use and possible side effects. In addition, pharmacy students need to be educated about the legal and regulatory entities governing MC use in different jurisdictions to comply with local laws and ethical standards. By incorporating comprehensive cannabis education into pharmacy curriculums, future pharmacists may be able to confidently contribute to evidence-based medicine and patient care to facilitate informed decision-making over MC use and RC use [11,14,15,16].

With the growing trend in state legalization and increased use of cannabis, pharmacy students and pharmacists are likely to be consulted by patients and other healthcare professionals on the safety, efficacy, and drug–drug interactions of cannabis. Furthermore, pharmacists are likely to be involved in policy developments and consulted on legal matters for the use of cannabis. With state policies and practices changing ever so rapidly, it is not unreasonable to assume pharmacists and other healthcare professionals will have to be capable of answering and making decisions regarding cannabis therapeutics and dispensations for patients. However, many pharmacy students believe they need more time to be ready to provide such a level of information to patients regarding safety, efficacy, and legal implications [17]. On the other hand, while current practicing pharmacists feel comfortable answering clinical questions, most would prefer additional education to ensure proper safety and efficacy standards are met [18]. The literature in this area primarily has a sample of a single pharmacy school or a few schools and is descriptive. This study aims to understand pharmacy students’ preparedness to counsel patients with cannabis use disorder (CUD) and evaluate the relationship between knowledge, attitudes, and behavioral intention to counsel over CUD.

## 2. Materials and Methods

### 2.1. Study Design and Population

A structured prospective cross-sectional survey was administered to professional pharmacy students in Accreditation Council for Pharmacy (ACPE)-accredited pharmacy schools across the USA with various levels of legalization of RC and MC within their respective states. The pharmacy students were either in the P1, P2, P3, or P4 years of their professional program at the time of administration of the survey. The survey was conducted between 1 October 2022 and 30 April 2023.

### 2.2. Sample Selection

A systematic sampling approach was utilized for the recruitment of participants. Inclusion criteria consisted of the student being at least 18 years old and a current pharmacy student attending an accredited college of pharmacy in the USA. Up to two follow-up reminder emails were sent to help with the recruitment. 

### 2.3. Instrument Design

A structured questionnaire with close-ended questions using predefined answers or a Likert scale was used to gather information about the sample characteristics, knowledge, attitudes towards MC use and RC use, and behavior intention to counsel over CUD. This survey was adapted from previous research looking to assess knowledge and attitude towards medical marijuana, which was then modified to fit the needs of this study [16]. The first section assessed knowledge about CUD. The second section included items on attitudes towards medical and recreational use of cannabis. The third section assessed behavioral intention toward counseling patients with CUD. The survey ended with the sample characteristics questions. 

The knowledge section contained ten items to characterize the participants’ overall understanding of CUD using a true/false scale. These ten items assessed the general knowledge, the diagnosis criteria, and the disease presentation of CUD, along with the pharmacokinetics of cannabis, cannabis intoxication diagnostic criteria, cannabis usage rates among the population, two items over symptoms of cannabis use, long-term side effects, and potential sources of risk factors amongst students. Each question in the knowledge section gave the student a true or false statement based on the current *DMS-V* guidelines over CUD. Each question was scored as 1 = correct or 0 = incorrect. After undergoing psychometric analysis, one item was removed due reliability analysis, yielding nine items that were used in the final analysis as per the study objectives.

The attitudes section contained 11 items assessing the participants’ attitudes towards the use of cannabis and its potential hazards in the medical and recreational settings using a 5-point Likert scale (1 = strongly disagree, 2 = disagree, 3 = neutral, 4 = agree, or 5 = strongly agree). This survey section contained two distinct domains: attitudes toward MC and RC use. The first domain, attitude towards medical cannabis (MC) use, had five items and aimed to determine the students’ position regarding medical cannabis use legality, safety when used responsibly, potential for abuse and criminal activity, professional knowledge, and primary source of support for cannabis use. The second domain, attitude towards recreational cannabis use, had the first five items identical to the first but now with focus on recreational cannabis use rather than medical. In addition, a sixth item captured the respondents’ perceptions of the adverse effects of cannabis use.

The behavioral intention of confidence in counseling on CUD section contained four items that aimed to understand behavioral intentions in real-life situations that would be expected of a pharmacist. The section aimed to determine the students’ behavioral intention based on their confidence in answering questions or concerns about cannabis use, safety, abuse potential, and referral for potential therapy for CUD. Like the attitudes section, a 5-point Likert scale was used in the behavioral intention section (1 = strongly disagree, 2 = disagree, 3 = neutral, 4 = agree, 5 = strongly agree). 

Lastly, sample characteristics included a section collecting gender, age, ethnicity, academic year, and other information specific to student understanding of the legality of cannabis use and its knowledge. Students were explicitly asked to report their awareness of the MC and RC legality in the state where they are actively attending pharmacy school. Past patient interaction looked to assess if the students had interacted with a patient using cannabis. Obtained knowledge asked students to report where they had gained most of their knowledge of cannabis used recreationally or medically.

### 2.4. Data Collection Process

This structured survey was created using Qualtrics software. Associate deans of pharmacy schools were sent a recruitment email to consider forwarding our survey to all their students within the college. Distribution amongst students was carried out by administrative staff within each college by emailing students attending their respective colleges of pharmacy. The survey was offered to participants; it could be taken either on tablets/laptops or personal mobile phones by providing the URL or a QR code. Participation was anonymous and voluntary. The survey included a consent letter in which every student choosing to participate in this study was required to provide consent by accepting the consent letter. Sample size calculation using the QualtricsXM sample size calculator indicated a sample of 380 to be adequate. The parameters required had a confidence interval of 95%, a total population size of 47,529 representing the entire student body, and a margin of error of 5% [19]. Completed surveys were downloaded, and data were stored on password-protected computers. To promote participation, once the participants completed the study survey, they were entered into a raffle through an optional document to include their name and email to win a USD 200 gift card that would be provided at the end of the recruitment period. The University Institutional Review Board approved this study. 

### 2.5. Data Analyses

An alpha of 0.05 was used to determine statistical significance across all tests. The collected data were analyzed using descriptive and inferential statistics with SAS version 9.4M7. Continuous variables were summarized using mean and standard deviation (SD), while categorical variables used frequency and percentages of the cohort study population. Independent sample *t*-test, chi-square test, Fisher’s exact, and one-way analysis of variance were conducted to assess variation in the study population and to identify any relationships between behavioral intention scores stratified by sample characteristic variables. The knowledge section score was represented by the average percent correct responses out of a total possible score of 100. The scores for attitudes towards MC use and RC use, along with the BIs, were the average of each respective section ranging from 1 to 5. To capture the internal consistency of the modified survey, Cronbach’s alpha was calculated for the attitudes and behavioral intentions sections. Bivariate Pearson correlation analyses examined the strength and directional relationships between knowledge, attitudes toward MC use, and attitudes towards RC use and BIs. Multivariate linear regression analysis was undertaken to identify factors associated with behavior intention based on knowledge, attitudes, and sample characteristics as the independent variables in the model.

## 3. Results

### 3.1. Sample Characteristics

A total of 513 potential students initiated the survey. Of these, 102 participants did not complete the survey with comprehensive responses, warranting exclusion from the analysis. With an overall response rate of 80.1%, the remaining 411 students who completed the survey were included in the final study analysis. The average age of respondents was 24.7 (SD = 4.2). Nearly three-quarters of the respondents were female (74.2%). The distribution between class ranks primarily consisted of first- and third-year students at 38.7% and 32.4%, respectively. The most prominent ethnic group was Caucasians (41.4%), followed by Asians (31.3%). Nearly three-quarters of the students (73.2%) resided in recreationally legal states, while being aware of MC and RC use in their state was 66.7% and 73.2% of the time, respectively. Nearly 4 in 10 (39.2%) students responding indicated they acquired their knowledge of cannabis through a seminar lecture outside of their pharmacy curriculum. While most students (51.3%) did have a past interaction with a patient using cannabis, there was not a significant difference in frequency (Table 1). Significant differences existed in behavioral intention scores by gender, ethnicity, academic year, awareness of MC, past patient interaction, and obtained knowledge. The study sample was comparative to the national statistics for pharmacy students by race and gender without any major deviation [19].

### 3.2. Knowledge, Attitudes, and Behavior Scores towards CUD

The students’ average score on the knowledge section was 81% (SD 16%), ranging from 11% to 100%. One question from this section was removed from the final analysis as further revision indicated an apparent ambiguity in this question’s interpretation. The final survey analysis was scored out of nine, where a perfect score would equate to 100% in the knowledge section. The resulting attitudes towards MC use in domain one and attitudes towards RC use in domain two resulted in an average sub-score of 4.13 (SD 0.75) and 3.28 (SD 0.80), respectively, with ranges from 1 to 5. Lastly, the average score for the behavioral intentions section was 2.74 (SD 1.00), ranging from 1 to 5. The Cronbach’s alpha of the modified survey measuring internal consistency for the attitudes towards MC, attitudes towards RC, and behavioral intentions sections were 0.80, 0.80, and 0.91, respectively (Table 2).

### 3.3. Relationship of Knowledge, Attitudes, and Behavioral Intentions towards CUD

The correlation of knowledge–attitudes toward MC use (r = 0.33, *p* < 0.0001) and knowledge–attitudes toward RC use (r = 0.12, *p* < 0.05) both were positively correlated. Attitudes towards MC use–attitudes toward RC use (r = 0.64, *p* < 0.0001) were positively correlated along with attitudes toward RC use–BIs (r = 0.22, *p* < 0.0001). Both knowledge-BIs and attitudes towards MC use–BIs trended negatively but were insignificant (Table 3).

Multivariate linear regression to predict factors associated with behavioral intension on confidence to counsel on CUD indicated attitudes towards MC use (β = −0.31, *p* < 0.0007), attitudes towards RC use (β = 0.43, *p* < 0.0001), academic year: P2 (β = −0.45, *p* < 0.0012), awareness of MC legality (β = 0.31, *p* < 0.0394), obtained knowledge: seminar lecture (β = −0.24, *p* < 0.0379), and past patient interaction (β = 0.24, *p* < 0.0245) were all significantly associated with behavioral intentions after controlling for confounders (Table 4).

## 4. Discussion

The presented results of this study showcased that despite adequate knowledge, there is an apparent gap in pharmacy students’ behavioral intention to counsel. Pharmacy students were reasonably knowledgeable regarding CUD, scoring above 80% on average. Their attitudes towards MC use were relatively high compared to a slight decrease in attitudes towards RC use, but overall positive. However, their behavioral intentions regarding their confidence in counseling for CUD were largely negative. The discrepancy between attitudes toward MC use and RC use might be associated due to a variety of reasons, such as a lack of regulatory control and clinical oversight of individuals using cannabis recreationally, the general demographic of those using cannabis recreationally, and, potentially, the lack of comfort to address appropriately and counsel those using cannabis recreationally. However, after controlling for age, gender, ethnicity, academic year, and awareness of the policies of MC and RC use in the students’ respective states, the multivariate linear regression indicated that pharmacy students with positive attitudes towards RC use, compared to MC use, were more likely to have the intent to counsel patients. Performance in knowledge scores and the overall positive attitudes towards MC use and RC use were consistent with the current literature and the low behavioral intentions scores’ association with a general lack of confidence in counseling patients [16,20]. This study showed similar results to other study cohorts reporting a desire for more CUD-related content to be incorporated into their respective curriculums [20]. In addition, behavioral intentions scores further varied based on most sample characteristics measured except on the awareness of RC legality and the legality in the respondent’s state. The students in this sample were more likely to obtain most of their knowledge of cannabis from non-pharmacy curriculum lectures or past recreational use before an accredited pharmacy curriculum lecture taught by appropriately distinguished educators and professors. A cause for concern rises, as many outside sources are not subjected to the validation or falsifiability in which healthcare curriculums founded on the evidence-based literature typically are. This inevitably raises the concern for spreading misinformation, further driving a gap in knowledge, attitudes, and BI. All of which brings to question how to standardize education and attain the necessary knowledge regarding cannabis therapeutics, safe administration, and management of potential adverse effects regardless of the student’s characteristics or experience with cannabis.

As for potential limitations, only a few pharmacy colleges responded, despite the survey being sent out to schools nationwide. In addition, the schools that responded were heavily biased toward Texas-based pharmacy schools. While this study takes a significant step in covering students’ perceptions at a large scale, to achieve generalizability with minimal bias, students from all geographical and societal backgrounds must be accounted for. A skewness in the distribution of the geographical location of responses will sway the outcomes in favor of the population in those respective locations. 

All the while, this study does display various strengths. This study highlights how a partial lack of knowledge about cannabis is associated with a corresponding lack of confidence in preparedness to council patients over CUD. The results of this study can help promote the need for additional training amongst students to tackle CUD counseling scenarios in the field. The results also propose the need for additional education and preparation on cannabis-related topics and patient counseling initiatives to help build students’ confidence in conveying the clinical information necessary to meet patient demands. Lastly, a significant number of surveys were incomplete, leading to the exclusion of potentially valuable data. While the complete survey responses provided valuable insights, the necessity to exclude incomplete data may have impacted the comprehensiveness of the findings.

Since the legalization of medical cannabis use in California (1996), 41 states have followed with legalization, along with 4 states allowing for CBD oil containing THC, while only 5 states that completely criminalize the use of cannabis medically or recreationally remain [21,22]. Perceptions of cannabis use amongst cannabis users and abstainers have varied widely but remained optimistic regarding its potential benefit and its management of possible harmful adverse effects. However, due to public information lacking in quantity and quality, the results and adverse effects of cannabis can easily be interpreted and misrepresented, calling for more attention to be placed on providing educational information for the public [23].

The current literature reports a general desire among healthcare trainees for increased training on knowledge of cannabis use to counsel patients. While previous studies have highlighted the benefits of integrating or further supplementing CUD education into pharmacy school curriculums, the findings are often limited to single or a small number of local pharmacy school cohorts. Therefore, there is a crucial need to expand research efforts to assess the overall readiness and behavioral intentions of pharmacy students on a comprehensive national scale. By doing so, a better understanding would be attained to address the broader education needs and perspectives of future pharmacists in counseling patients on CUD across diverse healthcare settings [24,25]. Based upon the domains of CUD knowledge identified in the existing literature such as pharmacology, therapeutic uses, legal considerations, communication skills, and patient-centered care, educators can develop tailored educational programs to equip current pharmacy students with the necessary knowledge base and capability to counsel patients on CUD [26,27]. Specifically, incorporating experiential learning opportunities, such as case-based learning and simulated patient encounters, can enhance students’ ability to apply theoretical knowledge in practical scenarios, thereby improving their readiness and confidence in counseling patients [28,29]. Moreover, effective communication skills training would facilitate open and non-judgmental discussions about cannabis use, inevitably fostering trust and rapport between pharmacists and patients, ultimately geared towards improving patient outcomes [30,31]. By further implementing these practices into pharmacy curriculums, institutions can better prepare future pharmacists to address the complex challenges and evolving landscape of cannabis use within various healthcare settings.

## 5. Conclusions

While the behavioral intention of pharmacy students to counsel patients over CUD was low, there was a significant positive association between behavioral intention and attitude toward recreational cannabis use. However, attitudes towards MC use did significantly lower the willingness to counsel. Furthermore, knowledge level amongst the students was not associated with the behavioral intention to counsel CUD. This lack of confidence points to a need for additional preparedness of students to be able to be proficient future pharmacists. In addition, preparation oriented around cannabis-based agents with medical and recreational intentions is needed. This would ensure that future pharmacists are adequately prepared to handle the evolving healthcare landscape and address the growing patient demand for information and guidance regarding cannabis use and associated CUD. Pharmacy schools can play a pivotal role in producing knowledgeable, confident, and capable healthcare professionals able to navigate the complexities of cannabis-related interventions and the needs of patients.

## Figures and Tables

**Table 1 pharmacy-12-00077-t001:** Sample characteristics and behavioral intentions on confidence to counsel over CUD (N = 411).

Variable	Characteristics	Frequency	%	Mean	SD	*p*-Value
Gender	Male	106	26	2.99	0.99	0.0027 #
	Female	305	74	2.65	0.99	
Ethnicity	Caucasian	170	41	2.67	0.95	0.0377 ^
	Black or African American	35	9	2.95	1.14	
	Hispanic or Latino	42	10	2.40	0.98	
	Asian	128	31	2.81	1.00	
	Native Hawaiian or Pacific Islander	4	1	2.31	1.55	
	Middle Eastern	32	8	3.06	0.97	
Academic Year	Pharmacy Student—P1	159	39	2.78	1.05	<0.0001 *
	Pharmacy Student—P2	70	17	2.27	0.82	
	Pharmacy Student—P3	133	32	2.85	1.00	
	Pharmacy Student—P4	49	12	2.98	0.92	
Legal status	No	110	27	2.72	1.03	0.4918 #
	Yes	301	73	2.79	0.94	
Awareness of MC legality	No	137	33	2.55	1.07	0.0112 #
	yes	274	67	2.83	0.95	
Awareness of RC legality	No	110	27	2.60	1.06	0.1093 #
	Yes	301	73	2.79	0.98	
Past patient interaction	No	211	51	2.64	1.02	0.0413 #
	Yes	200	49	2.84	0.98	
Obtained knowledge	Past recreational use	122	30	2.87	1.04	0.0341 *
	Past medical use	8	2	3.19	1.29	
	Curriculum lecture (PharmD curriculum)	120	29	2.80	0.98	
	Seminar lecture (outside of PharmD curriculum)	161	39	2.57	0.96	

All data are in n (%) format unless stated. MC = medical cannabis; RC = recreational cannabis. # = *t*-test; * = ANOVA test; ^ = Fisher’s exact test. Behavioral intentions were scored using a 5-point Likert scale (1 = strongly disagree, 2 = disagree, 3 = neutral, 4 = agree, or 5 = strongly agree).

**Table 2 pharmacy-12-00077-t002:** Pharmacy students’ knowledge, attitudes, and behavioral intentions towards CUD (N = 411).

	Mean	SD	Minimum	Maximum	Cronbach’s *α*
Knowledge (% correct)	81	16	11	100	
Attitudes towards MC use	4.13	0.75	1.00	5.00	0.80
Attitudes towards RC use	3.28	0.80	1.00	5.00	0.80
Behavioral intention	2.74	1.00	1.00	5.00	0.91

MC = medical cannabis; RC = recreational cannabis. A 5-point Likert scale (1 = strongly disagree, 2 = disagree, 3 = neutral, 4 = agree, or 5 = strongly agree) scored participants’ attitudes towards both MC and RC and their behavioral intentions.

**Table 3 pharmacy-12-00077-t003:** Pearson correlation coefficient for knowledge, attitude, and behavioral intension towards CUD (N = 411).

	Knowledge	Attitudes towards Medical Cannabis Use	Attitudes towards Recreational Cannabis Use
Knowledge	-		
Attitudes towards medical cannabis use	0.33 **	-	
Attitudes towards recreational cannabis use	0.12 *	0.64 **	-
Behavioral intentions	−0.09	−0.01	0.22 **

* Correlation is significant at the <0.05 level. ** Correlation is significant at the < 0.0001 level.

**Table 4 pharmacy-12-00077-t004:** Predictors of behavioral intention to counsel patients over CUD (N = 411).

Variables	Coefficient (β)	*p*-Value
Attitudes towards MC use	−0.31	0.0007
Attitudes towards RC use	0.43	<0.0001
Academic year enrolled: P2	−0.45	0.0012
Awareness of MC use legality	0.31	0.0394
Source of prior knowledge seminar lecture	−0.24	0.0379
Past patient interaction	0.24	0.0245

MC = medical cannabis; RC = recreational cannabis. Covariates listed are those that showed significance in linear regression. Covariates included in the analysis were attitudes towards MC use, RC use, knowledge, academic year enrolled, awareness of MC use legality, source of prior knowledge, ethnicity, past patient interaction, awareness of RC legality, and cannabis legality.

## Data Availability

The data presented in this study are available on request from the corresponding author due to privacy and ethical restrictions.

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
