# Peer review of "Understanding Pharmacy Students’ Preparedness towards Counseling over Cannabis Use Disorder"

_pharmacy, 2024, doi:10.3390/pharmacy12030077_

Round 1

Reviewer 1 Report (Previous Reviewer 3)

Comments and Suggestions for Authors

As stated in my original review on this paper in late December, the methodology is sufficiently flawed such that the results do not support the stated conclusions.  Specifically, the authors aim to determine student pharmacists' preparedness to counsel patients with CUD, and evaluate underlying knowledge, attitudes and intentions to counsel on CUD. The methods used included variability  such that the results are not interpretable. The authors conclude that based on this study, comprehensive curricular  change (on CUD), and state policy and practice changes should be emphasized. (lines 23-25).  This was not substantiated in your study. In order to determine if the curriculum has suitably prepared students for counseling (on CUD), one would need to know when/where in the curriculum (if at all) did the students receive education on the topic.  If students had not yet received their formal instruction on CUD, then the survey results will indicate that the curriculum did not prepare them...when in fact they had not yet received the instruction, and if they had, the outcomes of the survey would likely be different.  The authors chose not to change this aspect of their study design, citing a paper that indicates some P1 students receive instruction on CUD.  True, but the authors do not know if the students in their own survey were among these P1s who had already received instruction on CUD (and likewise did the P2s in their survey receive their instruction prior to completing the survey?). It makes little sense to me to include students in the survey if they have not yet received their formal training on CUD.  By including all comers (P1-P3), regardless of where in their curriculum, if at all, the CUD instruction occurred, constitutes flawed methodology, in terms of making concluding comments stating that formal college curriculums require revision to address the topic of CUD counseling preparedness, and in my view, even when assessing students' knowledge, attitudes and intent (per the study's aim). Clearly, students with formal instruction on the topic would have different views than those with any curricular exposure to the topic. I believe the authors need to know what % of their survey respondents had received formal training in CUD, and then a sub-analysis of the data could be presented to give a better representation of how well the curriculum is preparing them.  As is, the study's conclusions are not substantiated by this study.  Based on flawed methodology, I do not endorse publishing this paper.   

Comments on the Quality of English Language

Minor edits required. 

Author Response

  • Reviewer #1
    • As stated in my original review on this paper in late December, the methodology is sufficiently flawed such that the results do not support the stated conclusions.  Specifically, the authors aim to determine student pharmacists' preparedness to counsel patients with CUD, and evaluate underlying knowledge, attitudes, and intentions to counsel on CUD. The methods used included variability such that the results are not interpretable. The authors conclude that based on this study, comprehensive curricular change (on CUD), and state policy and practice changes should be emphasized. (lines 23-25).  This was not substantiated in your study.
      • Thank you for your suggestion, this portion was revised to focus on the modification of the education and preparation in pharmacy schools to better prepare students of future counseling scenarios. Additional references were added in the discussion to support this claim based on the results. It should be noted that the goal was to measure Knowledge and we did measure the year students were enrolled and controlled for year enrolled in our models.
    • To determine if the curriculum has suitably prepared students for counseling (on CUD), one would need to know when/where in the curriculum (if at all) did the students receive education on the topic.  If students had not yet received their formal instruction on CUD, then the survey results will indicate that the curriculum did not prepare them...when in fact they had not yet received the instruction, and if they had, the outcomes of the survey would likely be different. 
      • We understand the point of concern regarding this verbiage, we have gone back through the manuscript and removed any portion that mentions curriculums not preparing the students to modifying or supplementing the training and preparation even more so in the abstract, introduction, and discussion sections. It should be noted that we did measure if they had knowledge on the topic which does act as in general if they would need this information and we had adequate students at all levels of year enrolled.
    • The authors chose not to change this aspect of their study design, citing a paper that indicates some P1 students receive instruction on CUD.  True, but the authors do not know if the students in their own survey were among these P1s who had already received instruction on CUD (and likewise did the P2s in their survey receive their instruction prior to completing the survey?). It makes little sense to me to include students in the survey if they have not yet received their formal training on CUD. 
      • Thank you for the comment. We chose to measure the source of the student’s past knowledge of cannabis use which is shown in the table 1 and later controlled for it in the regression analysis. We did not ask a specific question on knowledge of CUD, as the goal was to test them on this using knowledge questionnaire. However, this is why we elected to perform a multivariate regression analysis with academic year (P1-P4) included in the model to provide its impact on behavioral intention while accounting for class standing which would include when they’ve received the education over CUD. After controlling for the knowledge aspect, we still see a significant difference in behavioral intention among the students. We have included this as a potential limitation for not using the specific terms.
    • By including all comers (P1-P3), regardless of where in their curriculum, if at all, the CUD instruction occurred, constitutes flawed methodology, in terms of making concluding comments stating that formal college curriculums require revision to address the topic of CUD counseling preparedness, and in my view, even when assessing students' knowledge, attitudes and intent (per the study's aim). Clearly, students with formal instruction on the topic would have different views than those with any curricular exposure to the topic. I believe the authors need to know what % of their survey respondents had received formal training in CUD, and then a sub-analysis of the data could be presented to give a better representation of how well the curriculum is preparing them.  As is, the study's conclusions are not substantiated by this study.  Based on flawed methodology, I do not endorse publishing this paper.   
      • Thank you for your feedback. We understand the importance of understanding when and where students received education on Cannabis Use Disorder (CUD) within their curriculum and its potential impact on their preparedness to counsel patients. We added further explaining in the discussion section regarding the controlling of variables such as academic year for the regression model. By controlling academic year (for the P1-P4 student classification) this should control for any effect that comes with the academic year of the student. This approach was intended to provide a robust output of behavioral intention scores, considering the varying levels of exposure to CUD education among students. We acknowledge the limitations of our study design and the potential for bias introduced by including students at different stages of their curriculum. However, the goal of this study was to evaluate behavioral intension. As such our analysis still provides valuable insights into the overall perceptions of students’ preparedness of pharmacy students to counsel patients on CUD.

Reviewer 2 Report (Previous Reviewer 2)

Comments and Suggestions for Authors

The authors have addressed an important goal of capturing evidence that a partial lack of knowledge about cannabis is associated with a corresponding lack of confidence in preparedness to council patients over cannabis use disorder. The implications for improving the pharmacy students’ curriculum to include more formal training in the area appears to be supported.  

The authors have more clearly described the survey and the results in the current version of the manuscript. 

A major concern to consider is the following.

As it the study describes a cross-sectional survey, cause and effect is not definitive. A cohort study in which knowledge was obtained or not obtained would need to be conducted to definitively make the conclusion that the lack of knowledge caused the corresponding lack of confidence to council. Descriptions of a definitive cause-and-effect nature of the relationship should be removed from the manuscript. 

Some minor points are the following.

In the first sentence of the abstract there appears to be a word missing between concerns and pharmacy. Also, the word “cause” may be removed as mentioned in the major concern described above. A suggested revision is the following. 

The rise in cannabis use prompts significant concerns regarding pharmacy students' abilities to counsel patients over cannabis use disorder. 

For the p-values shown in Table 1, it was not stated which test was being done. I am supposing that they are the results from one-way analysis of variance tests. If so, please indicate so or indicate which test was conducted in reference to the corresponding p-values listed. 

Comments on the Quality of English Language

Minor grammatical improvements are suggested. 

Author Response

  • Reviewer #2
    • The authors have addressed an important goal of capturing evidence that a partial lack of knowledge about cannabis is associated with a corresponding lack of confidence in preparedness to council patients over cannabis use disorder. The implications for improving the pharmacy students’ curriculum to include more formal training in the area appears to be supported. The authors have more clearly described the survey and the results in the current version of the manuscript. 
      • We thank the reviewer for these positive comments.
    • A major concern to consider is the following.
      • As it the study describes a cross-sectional survey, cause and effect is not definitive. A cohort study in which knowledge was obtained or not obtained would need to be conducted to definitively make the conclusion that the lack of knowledge caused the corresponding lack of confidence to council. Descriptions of a definitive cause-and-effect nature of the relationship should be removed from the manuscript. 
        • The text was modified to represent the association of these variables, rather than a cause-and-effect relationship, that could be ascertained in a cross-sectional survey-based study.
      • Some minor points are the following.
        • In the first sentence of the abstract there appears to be a word missing between concerns and pharmacy. Also, the word “cause” may be removed as mentioned in the major concern described above. A suggested revision is the following. 
          • The rise in cannabis use prompts significant concerns regarding pharmacy students' abilities to counsel patients over cannabis use disorder. 
        • For the p-values shown in Table 1, it was not stated which test was being done. I am supposing that they are the results from one-way analysis of variance tests. If so, please indicate so or indicate which test was conducted in reference to the corresponding p-values listed. 
          • Thank you for both suggestions, the manuscript was modified accordingly and revised with the provided information.

Reviewer 3 Report (New Reviewer)

Comments and Suggestions for Authors

The manuscript aims to presents students’ knowledge of, attitudes towards and as well as behavior intention to pharmaceutical counselling on the topic of cannabis use disorder. The manuscript focuses on distinguishing patterns of factors influencing ability to provide counsel on CAD. The presented results show that despite adequate knowledge, there is a gap in students’ behavioral intention to counsel. Authors described their chosen methodology, explained statistical approach to their collected data. Data presentation is mostly clear and consistent, the article is well written. Most literature is recent and on the topic. The aim of the study is clearly stated and the conclusions are consistent with the presented evidence.

I would like to raise some points that could be better addressed by authors:

1)      The authors addressed an important topic, i.e. the ability to provide information by future pharmacists. While the aim of the study is clear, a research gap is not clearly stated. The authors should declare, what was missed or overlooked in previously published research. Because the topis has been approached numerous times in last 5 years. Especially, that they used a similar approach to the study from 2015. What was the initial hypothesis of the study?

2)      The manuscript lacks a section to describe implications for practice, which could be valuable for readers of Pharmacy. The authors wrote that gaps in knowledge should be properly addressed by pharmacy schools. Which domains of knowledge are in need to be addressed? Especially that there already exists a body of literature on education on cannabis: https://doi.org/10.1177/17151635211041041, https://doi.org/10.1016/j.ctim.2021.102675. In line 285 authors suggest that there curricula should be enriched with methods addressing counselling. What methods could authors recommend specifically? Without addressing these issues, manuscript may be of little interest to the readers.

3)      The authors wrote in line 258, that the study showed attitudes and knowledge of large sample of students with, among other, various political backgrounds. Firstly, the sample turned out to be just above the statistically relevant threshold. Secondly, the survey did not assess the political attitudes of students.

4)      It could also be interesting to see how do the attitudes of these students who chose to answer relate to average pharmacy students. Were they more likely to use cannabis for recreation or did the sample was representative to average student in this regard? In other words, did authors receive answers from

5)    The authors did not really discuss their findings with other, similar surveys. And there have been numerous attempts to measure knowledge and attitude on medical and recreational cannabis, e.g.: https://doi.org/10.5688/ajpe6296, https://doi.org/10.1016/j.japh.2019.08.008, The list of cited literature should include more studies similar to the manuscript.

6)      Authors made some choices that were never explained. Firstly, they excluded 1/5 of surveys, secondly they excluded one ambiguous question from questionnaire. The reason for these decisions should be better explained.

7)      Study limitations should be clearly stated and commented. One such limitation is the fact, that authors excluded vast majority of publications, due to availability of full texts.  

I think that the manuscript should be revised in order to be more meaningful and better discuss the results with previously published findings.

Author Response

  • Reviewer #3
    • The manuscript aims to presents students’ knowledge of attitudes towards and as well as behavior intention to pharmaceutical counselling on the topic of cannabis use disorder. The manuscript focuses on distinguishing patterns of factors influencing ability to provide counsel on CAD. The presented results show that despite adequate knowledge, there is a gap in students’ behavioral intention to counsel. Authors described their chosen methodology, explained statistical approach to their collected data. Data presentation is mostly clear and consistent, the article is well written. Most literature is recent and on the topic. The aim of the study is clearly stated, and the conclusions are consistent with the presented evidence.
      • We thank the reviewer for these positive comments.

  • I would like to raise some points that could be better addressed by authors:
    • The authors addressed an important topic, i.e. the ability to provide information by future pharmacists. While the aim of the study is clear, a research gap is not clearly stated. The authors should declare, what was missed or overlooked in previously published research. Because the topis has been approached numerous times in last 5 years. Especially, that they used a similar approach to the study from 2015. What was the initial hypothesis of the study?
      • Thank you for the suggestion, we’ve gone back and added in the appropriate references in the discussion to literature on the topic and specifically addressing the gap in literature and how our study seeks to help bridge that gap from lines 288-309.
    • The manuscript lacks a section to describe implications for practice, which could be valuable for readers of Pharmacy. The authors wrote that gaps in knowledge should be properly addressed by pharmacy schools. Which domains of knowledge are in need to be addressed? Especially that there already exists a body of literature on education on cannabis: https://doi.org/10.1177/17151635211041041, https://doi.org/10.1016/j.ctim.2021.102675. In line 285 authors suggest that there curricula should be enriched with methods addressing counselling. What methods could authors recommend specifically? Without addressing these issues, manuscript may be of little interest to the readers.
      • Thank you for the suggestion, we have addressed the concern by including a section on implications for practice in our manuscript in the discussion portion. Drawing upon existing literature on education on cannabis, including the references provided, we have identified specific domains of knowledge that are pertinent to counseling patients on CUD, such as pharmacology, therapeutic uses, legal considerations, communication skills, and patient-centered care. This information is now included.
    • The authors wrote in line 258, that the study showed attitudes and knowledge of large sample of students with, among other, various political backgrounds. Firstly, the sample turned out to be just above the statistically relevant threshold. Secondly, the survey did not assess the political attitudes of students.
      • Thank you for the suggestion, the text was edited accordingly to remove any mention of political implication.
    • It could also be interesting to see how do the attitudes of these students who chose to answer relate to average pharmacy students. Were they more likely to use cannabis for recreation or did the sample was representative to average student in this regard? In other words, did authors receive answers from
      • Thank you for the comment, we have added in text at lines 193-195 discussing how we ran a comparison of our study sample to the national pharmacy student using the data that can be accessed via the link provided below. We found no significant difference between age and race and other available characteristic variables.
    • The authors did not really discuss their findings with other, similar surveys. And there have been numerous attempts to measure knowledge and attitude on medical and recreational cannabis, e.g.: https://doi.org/10.5688/ajpe6296, https://doi.org/10.1016/j.japh.2019.08.008, The list of cited literature should include more studies similar to the manuscript.
      • Thank you for the suggestion, the above literature was referenced in the discussion section at lines 239-244. Due to word limitations originally, we elected to keep the discussion relatively concise. However, we’ve expanded this portion in more detail to help discuss the findings with the relevant literature.
    • Authors made some choices that were never explained. Firstly, they excluded 1/5 of surveys, secondly they excluded one ambiguous question from questionnaire. The reason for these decisions should be better explained.
      • The surveys excluded were those that were only partially completed, see lines 181-182. As stated in the data collection process section, only completed surveys were used for analysis. The reason for exclusion of the single questions was upon doing reliability analyses of the results, we realized the question was worded in an improper way which confused the survey takers and provided inappropriate and unreliable data. Based on interitem reliability we had to eliminate this item.  
    • Study limitations should be clearly stated and commented. One such limitation is the fact, that authors excluded vast majority of publications, due to availability of full texts.  
      • Thank you for the suggestion, we have gone back and added the references in the discussion section to provide the current literature on the topic based on an extensive literature search including, Moeller et. al., Caliguiri et. al., Kvillemo et. al., Zolotov et. al., and Parihar et. al.

  • I think that the manuscript should be revised in order to be more meaningful and better discuss the results with previously published findings.
    • Thank for the comment, we went ahead and revised as suggested based on all preceding comments and suggestions to improve the manuscript.

Round 2

Reviewer 1 Report (Previous Reviewer 3)

Comments and Suggestions for Authors

I appreciate that the authors took a look at my previous comments, and attempted to address the concerns.  However, I remain steadfast in stating that the methodology is flawed such that the conclusions are not justified.  I am likely repeating myself from the prior 2 reviews; the basic issue is that this paper surveyed pharmacy students in P1-P4, without asking them if/when/where in the curriculum did they receive instruction on CUD.  Without this piece of information/data, the authors cannot conclude that enhancements in the pharmacy curriculum are needed (lines 26-28; 247-248; 273-274; 320-321; 328-330).  Your conclusions are not justified based on your flawed methodology.  The authors state in lines 253-255 that the students in this sample were more likely to obtain most of their knowledge about CUD from non-pharmacy curriculum lectures or past recreational use (RC) (see also Table 1). Thus, the survey conducted did little to nothing to uncover whether or not current pharmacy curricula are useful in CUD education. Further, it is unclear if that was the authors intent in the first place, to study to quality of the curriculum in terms of preparing students for CUD in practice? If the authors intended to fill a gap in the literature, as stated in lines 80-81, they first need to identify a notable gap.  They state on lines 76-77 that many pharmacy students believe they need more time to be ready to effectively provide [CUD counseling], yet this study does not address what was covered at all in the pharmacy curriculum, so clearly cannot even begin to address whether "more time" is needed in a curriculum of study.  They go on to state in lines 78-79 that current pharmacy practitioners also prefer additional education to prepare them to properly counsel CUD patients. What additional education?  Might that be the gap you are trying to fill?  (Methods do not support that gap, as nothing on curriculum is covered in the survey). Thus, it is unclear what the gap was/is that the authors are trying to fill, by "aiming to understand pharmacy students' preparedness to counsel patients with CUD, and evaluate the relationship between knowledge, attitudes and BI to counsel on CUD".  Putting aside what gap is trying to be filled, the above-mentioned lack of analysis of the curriculum that the students completed is the major problem here. It would make sense that you would survey graduating P4 students to determine what instruction they received in the curriculum (including seeing patients (presumably during P4) which was a question on your survey).  Additional issues with this paper include the low number of respondents 411, out of >45K pharmacy students in the nation.. Also, 159 of the 411 were P1 students who likely had zero exposure to CUD in the curriculum yet, so why include them? Think of a clinical analogy: congestive heart failure clinic: Would you survey P1 students who had not yet studied heart failure and the pharmacotherapy to treat it, and have not yet seen patients in the clinic setting, who are just starting to learn patient communication? Would you ask them about their knowledge of heart failure, and their intentions to counsel these patients? Likely no. It does not make sense to me -it is premature to ask students about an outcome that they are not prepared to answer.  That is why I have suggested all along that this paper should include P4s (or preferably end of P4, to garner input from the P4 year in terms of preparing students to counsel once they are licensed). Take a look at Discussion, lines 234-235: "However, their BIs regarding their confidence in counseling for CUD were largely negative." The explanation for this is that the study methodology is flawed.  You included students whom the majority stated on your survey that they learned about CUD outside of the pharmacy curriculum or through their own personal RC... thus, while these students are in a pharmacy program, they are not yet qualified to give opinion on CUD curriculum in their college of pharmacy because they haven't received the instruction yet. In summary, based on my above comments, while I applaud your interest and work on an important clinical topic,  in which curricular enhancement could well be warranted, (though not proven in this paper),  I do not support publishing this paper.

Comments on the Quality of English Language

Some editing necessary.  

Author Response

Reviewer 3 Report (New Reviewer)

Comments and Suggestions for Authors

I believe that the authors addressed the suggestions and corrections I had previously proposed for the submitted manuscript. After reviewing the revised manuscript, I am pleased to report that I have no further comments or concerns. The modifications authors have made have effectively addressed all the points raised during the review process.

Author Response

We sincerely appreciate your thorough review of our manuscript and your positive feedback regarding the revisions we made. We are glad to hear that you found our responses to your suggestions and corrections satisfactory. Your feedback has been invaluable in improving the quality and clarity of our manuscript. Thank you for your time and attention to our work!

This manuscript is a resubmission of an earlier submission. The following is a list of the peer review reports and author responses from that submission.

Round 1

Reviewer 1 Report

Comments and Suggestions for Authors

This is a relevant topic for pharmacy education.

Some points of clarification are needed:

Line 54 – more information is needed on cannabis use disorder.

Throughout the paper – the use of acronyms is bothersome – I found myself searching throughout the paper for what some acronyms meant, such as BI, MC. The authors should spell the complete term the first time and put the acronym in brackets right after the complete spelling, after which the acronym should be used. The list of abbreviations at the end of the article comes too late.

Line 84 – the term “understand” is not appropriate. This study can gather information, but not achieve understanding. So, the aim should be about “gaining knowledge”, as the survey items did not provide understanding. Understanding is an act of cognition, knowledge is a different form of cognition, a basic form of cognition and usually needed to gain understanding.

Line 113 – same comment, “the knowledge section… characterize overall knowledge”, not understanding

Line 132 – I think validity would be strengthened about BI results if there were more items. This could be thought of as a limitation.

Line 224 – What criteria were used to conclude that pharmacy students were “reasonably” knowledgeable regarding CUD? What does reasonable mean? The term “acceptably” would garner the same question? How did the authors decide on a reasonable expectation?

Line 241 – the concern about outside sources not validated or scrutinized at the level of healthcare curriculum – more specifics are needed that indicate the assumption that the literature and practice are “correct’; otherwise we are assuming that because something is in the curriculum that it is “true”.

Discussion section – Much of the literature should be in the introduction section so the reader can gain some knowledge of the problem and purpose of the research. While the discussion should return to the literature for support, it should not be replete with new literature.

Limitations – clearly written, could also include limitation I already mentioned.

Conclusion – clearly articulated

Comments on the Quality of English Language

There are a few instances where word choice needs to be checked. For example, data is plural, so "data are" or "data were" versus data is or was. 

Check tense throughout, sometimes present may have been used where past tense was appropriate. 

Too much use of acronyms, for example, line 306 - the BI use...over CUD was low. Unless word count influenced the use of acronyms, in some cases spelling the words strengthens clarity of writing.

Reviewer 2 Report

Comments and Suggestions for Authors

The study describes the implementation of a multipart questionnaire in which correlations were sought across the designated sections. The correlations have relatively low correlation coefficients, and the significance is questionable for some of the comparisons. More information with the actual labeled scatter plots of each correlation should be provided. Although I could find the original study with most of the questions, the entire set of questions in the entire questionnaire should be available to the reader for review as part of the supplementary material. Specific questions and their contributions to the correlations where possible should be indicated. Currently the study appears to be abstract. More specifics regarding the correlations would benefit the reader and provide support for the conclusions. 

For example, for the section the authors labeled behavioral intention, the first question is regarding knowledge of subject about medical marijuana, and it is not about the behavioral intentions of the subject. If this is the question that shows a high correlation, then the correlation may not be due to the intended behavior of the subject. 

When using any previously questionnaire the precise changes that are made should be clearly indicated. 

As multiple correlations were conducted with the same dataset, a correction to the p-value would be expected using a Bonferroni correction or something comparable. 

The current study should do more to distinguish itself from the previous study by Moeller and Woods. More information on which questions are different should be provided as mentioned above. 

Comments on the Quality of English Language

There appeared to be some run-on sentences. 

Reviewer 3 Report

Comments and Suggestions for Authors

The methodology of this project is sufficiently flawed such that the results do not support the stated conclusions.  Specifically, you aim to determine student pharmacists' preparedness to counsel patients with CUD, and evaluate underlying knowledge, attitudes and intentions to counsel on CUD. The methods that you used were so broad and included such variability (comparing apples to oranges) that the results are not interpretable. You conclude that based on this study, curricular change (on CUD) is warranted, which has not been substantiated in your study. It would seem that in order to determine if the curriculum has suitably prepared students for counseling (on CUD), one would need to know when/where in the curriculum (if at all) did the students receive education on the topic, and what colleges of pharmacy are they from. It appears that this question was not posed. You seemed surprised (based on discussion comment Line 238) that the students learned about CUD in seminars outside of the curriculum, yet it is likely that the P1 students had not yet had the topic in their curriculum.  It seems logical to me that P1 students, and possibly P2 & P3 students, should have been excluded from your study, as the survey was administered as early as October of the academic year.  Thus, unless P1 students had received instruction on this topic in September of P1 (seems highly unlikely), or anytime during P1, if the student was a P2 student, then the student would be responding on your survey that they learned nothing about this topic in the curriculum. However, that doesn't mean that the curriculum was necessarily lacking in CUD content, but instead, the students just hadn't gotten to it yet (as 159 students in the sample were P1 students, who may receive instruction on this topic when they are P3s). Thus, by including P1-P3 students, you introduced significant variability which critically affected your interpretation of the results.  My suggestion: if you wish to determine how well the curriculum prepared students to counsel on CUD, it would make sense to include only P4 students (or the equivalent, if students were in an accelerated 3-year program). In that way, all students in the sample would have completed their didactic curriculum (comparing apples to apples), and thus you could determine the preparedness to counsel on the topic. You also state in limitations that your results are not generalizable as they did not include a broad sampling of students across the country, with most being from Texas.  The results did not include this distribution for the reader (what colleges participated), which is another shortcoming in the study.  Perhaps the study should have focused just on P4 students in colleges of pharmacy in Texas (you had 49 P4 students  - a very small sample size), which would yield students from more than one university, and all of those students would have the same cannabis laws in place (another source of variability from state to state).   In terms of students lack of confidence in counseling, this is a well-known shortcoming of pharmacy curricula that in terms of pre-APPE readiness, students are lacking in counseling skills, regardless of the topic, and this is not unique to CUD counseling; this should be expounded upon in the discussion (see VanLangen, K.M.; Schmidt, K.J.; Sohn, M.; Meny, L.; Bright, D.R. Development and initial evaluation of an Advanced Pharmacy Practice Experience readiness assessment plan. Am J Pharm Educ. 2023, 87, Article 9002. DOI: https://doi.org/10.5688/ajpe9002).

Comments on the Quality of English Language

Title - change 'over' to 'on'

Line 47 - suggest change 'especially' to 'including'

Line 51 - not sure comparison to cocaine and opiate stats is necessary

Line 52 - "federally illegal" - confusing because so many states have legalized RC use - reword 

Line 92 Methods - ACPE spelled out is missing Education

Line 118 - typo on DSM